

# Trend-preserving bias adjustment and statistical downscaling with ISIMIP3BASD (v1.0)

Stefan Lange[1]

[1]Potsdam Institute for Climate Impact Research (PIK), Member of the Leibniz Association, P.O. Box 60 12 03, 14412 Potsdam, Germany

**Correspondence:** Stefan Lange (slange@pik-potsdam.de)

**Abstract.** In this paper I present new methods for bias adjustment and statistical downscaling that are tailored to the requirements of the Inter-Sectoral Impact Model Intercomparison Project (ISIMIP). In comparison to their predecessors, the new methods allow for a more robust bias adjustment of extreme values, preserve trends more accurately across quantiles, and facilitate a clearer separation of bias adjustment and statistical downscaling. The new statistical downscaling method is

stochastic and better at adjusting spatial variability than the old interpolation method. Improvements in bias adjustment and trend preservation are demonstrated in a cross-validation framework.

## 1  Introduction

Bias adjustment in climate research is the adjustment of statistics of climate simulation data to the end of making them more

similar to climate observation data. In many application cases, these climate simulation and observation data have different spatial resolution. In most of these cases, the climate observation data are more highly resolved. In any of these cases, bias adjustment requires bridging the resolution gap.

In previous phases of the Inter-Sectoral Impact Model Intercomparison Project (ISIMIP; Warszawski et al., 2014; Frieler et al., 2017), climate simulation data were always more coarsely resolved than the climate observation data used for their bias

adjustment, and the goal of this bias adjustment was not just to remove systematic biases from the simulation data but also to increase their spatial resolution to that of the observation data. In application cases like these, bias adjustment as it is commonly understood involves two distinct problems, (i) the actual bias adjustment at the spatial resolution of the simulation data, and (ii) a statistical downscaling to the spatial resolution of the observation data.

Commonly, the bulk of resources for the development of solutions to these problems is allocated to problem (i), and problem

(ii) is solved by a mere spatial interpolation of the simulation data to the spatial resolution of the observation data prior to bias adjustment. For example, this approach was adopted in the ISIMIP Fast Track (Hempel et al., 2013), in ISIMIP2b (Frieler et al., 2017), and for the generation of the NASA Earth Exchange Global Daily Downscaled Projections data set (NEX-GDDP;



Thrasher et al., 2012). The simplicity of this approach comes at a price if, as usual, the same univariate bias adjustment method is independently applied in every cell of the observation data grid. The bias adjustment then retains the too high spatial coherence of the interpolated simulation data, and inflates temporal variability at their original spatial resolution (Maraun, 2013).

These issues can be overcome by spatially multivariate bias adjustment or, as suggested by Maraun (2013), using a statistical downscaling method which is able to add the spatiotemporal variability that is missing at the simulation data resolution. He argues that such a method should be stochastic, given the multivalued nature of statistical downscaling (there are infinitely many high-resolution fields compatible with the same low-resolution field) and the multifaceted inflation issues caused by deterministic methods such as spatial interpolation.

In this paper, I present the bias adjustment and statistical downscaling methods to be used in phase 3 of ISIMIP. These methods have been developed following the paradigm of a clear separation of bias adjustment and statistical downscaling. In ISIMIP3, climate simulation data shall first be bias-adjusted at their original spatial resolution using spatially aggregated climate observation data. In a second step, their spatial resolution shall be increased using the original climate observation data and a stochastic statistical downscaling method.

Next to this paradigm shift, the new bias adjustment method has been developed to work better than its predecessor in several respects. The following design decisions were taken in this context. While structurally different bias adjustment methods were used for different climate variables in ISIMIP2b (Frieler et al., 2017), the ISIMIP3 method applies a newly developed quantile mapping method to all climate variables since this allows for the controlled adjustment of biases in all quantiles. The new method is approximately trend-preserving in all quantiles and therefore features a more comprehensive trend preservation than

the ISIMIP2b method. The new quantile mapping method is parametric because this promises a more robust adjustment of biases in extreme quantiles than non-parametric quantile mapping (Switanek et al., 2017). The new bias adjustment method also includes a modified version of the event likelihood adjustment introduced by Switanek et al. (2017). This new feature facilitates a confinement of extreme values to the physically plausible range, which had to be enforced using cap values in ISIMIP2b.

The remainder of this paper is organized as follows. Climate simulation and observation data used in this study are described in Sect. 2. Details of the ISIMIP2b and ISIMIP3 bias adjustment and statistical downscaling methods are presented in Sect. 3. Also in Sect. 3 I explain how the new and old methods are tested in the following. Test results are presented and compared in Sect. 4. Conclusions are made in Sect. 5.

## 2   Data

### 30  2.1   Climate simulation data

Climate simulation data are taken from the fifth phase of the Coupled Model Intercomparison Project (CMIP5; Taylor et al., 2011). I use data produced with the four climate models that were also used in ISIMIP2b (GFDL-ESM2M, HadGEM2-ES, IPSL-CM5A-LR, MIROC5; Frieler et al., 2017). For bias adjustment at $2°$ spatial resolution, daily data for ten variables (see



**Table 1.** Climate variables considered in this study.

| Variable | Short name | Unit |
|---|---|---|
| Daily mean near-surface relative humidity | hurs | % |
| Daily mean precipitation | pr | $\mathrm{kg\,m^{-2}\,s^{-1}}$ |
| Daily mean snowfall flux | prsn | $\mathrm{kg\,m^{-2}\,s^{-1}}$ |
| Daily mean sea-level pressure | psl | Pa |
| Daily mean surface downwelling longwave radiation | rlds | $\mathrm{W\,m^{-2}}$ |
| Daily mean surface downwelling shortwave radiation | rsds | $\mathrm{W\,m^{-2}}$ |
| Daily mean near-surface wind speed | sfcWind | $\mathrm{m\,s^{-1}}$ |
| Daily mean near-surface air temperature | tas | K |
| Daily maximum near-surface air temperature | tasmax | K |
| Daily minimum near-surface air temperature | tasmin | K |

Table 1) are conservatively interpolated (Jones, 1999) to a global $2° \times 2°$ latitude–longitude grid. For bias adjustment at $1/2°$ spatial resolution, these data are bilinearly interpolated to a global $1/2° \times 1/2°$ latitude–longitude grid. I concatenate output data of the historical CMIP5 experiment with output data of the rcp85 CMIP5 experiment to obtain climate simulation data representing the historical time period 1980–2015. Only output data of the rcp85 CMIP5 experiment are used to obtain climate

simulation data representing the future time period 2064–2099.

## 2.2   Climate observation data

As observational reference data for bias adjustment and statistical downscaling I use the EartH2Observe, WFDEI and ERA-Interim data Merged and Bias-corrected for ISIMIP (EWEMBI; Lange, 2016), which cover the entire globe at $1/2°$ spatial and daily temporal resolution from 1979 to 2016. For statistical downscaling from $2°$ to $1°$ and for bias adjustment at $2°$ spatial

resolution, these data are conservatively aggregated to global $1° \times 1°$ and $2° \times 2°$ latitude–longitude grids, respectively.

## 3   Methods

### 3.1   ISIMIP2b method

The ISIMIP2b bias adjustment and statistical downscaling method is comprehensively described in Frieler et al. (2017), Lange (2018), and Hempel et al. (2013). For statistical downscaling, simulation data are bilinearly interpolated to the observation data

grid. These interpolated data are then bias-adjusted in different ways for different climate variables.

For pr, psl, rlds, sfcWind, and tas, monthly mean values are adjusted to the end of removing the bias in their historical multi-year mean value. This adjustment is done multiplicatively for pr, rlds, and sfcWind, and additively for psl and tas. In



order to preserve trends in multi-year monthly mean values, the same scaling factor respectively offset is used in all application periods. In a second step, day-to-day variability around the monthly mean value is adjusted using transfer functions derived for every calendar month from historical simulations and observations.

An indirect bias adjustment of tasmax and tasmin is done by adjusting $\mathrm{tasmax} - \mathrm{tas}$ and $\mathrm{tas} - \mathrm{tasmin}$ using monthly scaling factors which remove the bias in the mean value of all historical daily values of these non-negative variables from a given calendar month. The adjusted values are then added to and subtracted from bias-adjusted tas values in order to obtain bias-adjusted tasmax and tasmin values, respectively.

Bias adjustment of rsds is done by parametric quantile mapping using beta distributions with lower bounds of zero and upper bounds estimated by rescaled climatologies of downwelling shortwave radiation at the top of the atmosphere. For trend preservation, upper bounds, mean values, and variances of historical observations are modified using simulated trends prior to quantile mapping. Using beta distributions with fixed lower and upper bounds of 0 and 100 %, respectively, this method is also used to bias-adjust hurs. Bias-adjusted prsn values are obtained by multiplying bias-adjusted pr values with the original prsn-over-pr ratio. This ratio is therefore not bias-adjusted.

## 3.2 ISIMIP3 method

The newly developed ISIMIP3 bias adjustment and statistical downscaling method is comprehensively described in the following. It consists of a bias adjustment method that is applied at the spatial resolution of the climate simulation data and a statistical downscaling method that is applied to the bias-adjusted climate simulation data to the end of increasing its spatial resolution to that of the climate observation data. These two new methods are presented in the following two subsections.

### 3.2.1 ISIMIP3 bias adjustment method

The ISIMIP3 bias adjustment method is a parametric quantile mapping method that has been designed to (i) robustly adjust biases in all percentiles of a distribution and (ii) preserve trends in these percentiles. It is applicable for bias adjustment of different kinds of climate variables including those listed in Table 1. Like the ISIMIP2b bias adjustment method, it is independently applied to every variable, grid cell, and calendar month.

In order to overcome the zoo of approaches to bias adjustment used for different variables in ISIMIP2b, the new method features a unified framework, which can be specified for an application to one particular climate variable. Specifications for the variables considered here are listed in Table 2. Note that biases in prsn, tasmax, and tasmin are not adjusted directly. Instead, I adjust biases in pr and $\mathrm{prsnratio} = \mathrm{prsn}/\mathrm{pr}$, and multiply the resulting values to arrive at bias-adjusted prsn values. I do this in order to (i) ensure $0 \leq \mathrm{prsnratio} \leq 1$, and (ii) preserve trends in prsnratio. For tasmax and tasmin, Piani et al. (2010) point out that an independent bias adjustment of tas, tasmax, and tasmin may result in large relative errors in the daily temperature range, $\mathrm{tasrange} = \mathrm{tasmax} - \mathrm{tasmin}$, and the skewness of the daily temperature cycle, $\mathrm{tasskew} = (\mathrm{tas} - \mathrm{tasmin})/\mathrm{tasrange}$. They also demonstrate that these errors can be minimized by a direct and independent bias adjustment of tas, tasrange, and tasskew. Here, I follow their lead and derive bias-adjusted tasmax and tasmin values from bias-adjusted tas, tasrange, and tasskew values.





**Table 2.** Bias adjustment specifications for climate variables considered in this study. Note that the units of $\mathrm{prsnratio} = \mathrm{prsn/pr}$, $\mathrm{tasrange} = \mathrm{tasmax} - \mathrm{tasmin}$, and $\mathrm{tasskew} = (\mathrm{tas} - \mathrm{tasmin})/\mathrm{tasrange}$ are 1, K, and 1, respectively. For units of the other climate variables see Table 1. Note that the lower threshold of pr is equivalent to 0.1 mm/day.

| Variable short name | Lower bound | Lower threshold | Upper bound | Upper threshold | Distribution | Trend preservation | Detrending | Other |
|---|---|---|---|---|---|---|---|---|
| hurs | 0 | 0.01 | 100 | 99.99 | beta | bounded | no | – |
| pr | 0 | 0.1/86400 | – | – | gamma | mixed | no | – |
| prsnratio | 0 | 0.0001 | 1 | 0.9999 | beta | bounded | no | sampling of missing values |
| psl | – | – | – | – | normal | additive | yes | – |
| rlds | – | – | – | – | normal | additive | yes | – |
| rsds | 0 | 0.0001 | 1 | 0.9999 | beta | bounded | no | upper bound scaling |
| sfcWind | 0 | 0.01 | – | – | Weibull | mixed | no | – |
| tas | – | – | – | – | normal | additive | yes | event likelihood not adjusted |
| tasrange | 0 | 0.01 | – | – | Rice | mixed | no | – |
| tasskew | 0 | 0.0001 | 1 | 0.9999 | beta | bounded | no | – |

In the following, I will describe the unified framework of the ISIMIP3 bias adjustment method. In this context, let $x_{\mathrm{hist}}^{\mathrm{obs}}$ be the time series of historical observations for one climate variable, grid cell, and calendar month. Let further $x_{\mathrm{hist}}^{\mathrm{sim}}$ and $x_{\mathrm{fut}}^{\mathrm{sim}}$ be the simulated time series for the historical and future time period, respectively, and the same climate variable, grid cell, and calendar month. Since the bias adjustment method is trained on $x_{\mathrm{hist}}^{\mathrm{obs}}$ and $x_{\mathrm{hist}}^{\mathrm{sim}}$ the historical time period is also called the

training period. Since it is applied to $x_{\mathrm{fut}}^{\mathrm{sim}}$ the future time period is also called the application period.

The bias adjustment algorithm with inputs $x_{\mathrm{hist}}^{\mathrm{obs}}$, $x_{\mathrm{hist}}^{\mathrm{sim}}$, $x_{\mathrm{fut}}^{\mathrm{sim}}$ and output $y_{\mathrm{fut}}^{\mathrm{sim}}$ proceeds in the following steps, which are explained in more detail below.

1. (For rsds only.) Scale values in $x_{\mathrm{hist}}^{\mathrm{obs}}$, $x_{\mathrm{hist}}^{\mathrm{sim}}$, $x_{\mathrm{fut}}^{\mathrm{sim}}$ to the interval $[0, 1]$.

2. (For prsnratio only.) Replace missing values in $x_{\mathrm{hist}}^{\mathrm{obs}}$, $x_{\mathrm{hist}}^{\mathrm{sim}}$, $x_{\mathrm{fut}}^{\mathrm{sim}}$ by random sampling from available values.

3. (For psl, rlds, and tas only.) Detrend $x_{\mathrm{hist}}^{\mathrm{obs}}$, $x_{\mathrm{hist}}^{\mathrm{sim}}$, $x_{\mathrm{fut}}^{\mathrm{sim}}$.

4. (For bounded variables only.) Randomize values beyond threshold in $x_{\mathrm{hist}}^{\mathrm{obs}}$, $x_{\mathrm{hist}}^{\mathrm{sim}}$, $x_{\mathrm{fut}}^{\mathrm{sim}}$.

5. (For all variables.) Transfer the simulated climate change signal for every distribution quantile from $x_{\mathrm{hist}}^{\mathrm{sim}}$, $x_{\mathrm{fut}}^{\mathrm{sim}}$ to $x_{\mathrm{hist}}^{\mathrm{obs}}$. Let $x_{\mathrm{fut}}^{\mathrm{obs}}$ be the resulting time series of pseudo future observations.

6. (For all variables.) Use parametric quantile mapping to adjust the distribution of values in $x_{\mathrm{fut}}^{\mathrm{sim}}$ to the distribution of

values in $x_{\mathrm{fut}}^{\mathrm{obs}}$. For bounded variables, also bias-adjust the frequency of values beyond threshold. Let $y_{\mathrm{fut}}^{\mathrm{sim}}$ be the resulting time series.



7. (For psl, rlds, and tas only.) Add trend subtracted from $x_{\mathrm{fut}}^{\mathrm{sim}}$ in step 3 to $y_{\mathrm{fut}}^{\mathrm{sim}}$.

8. (For rsds only.) Scale values in $y_{\mathrm{fut}}^{\mathrm{sim}}$ back to their actual range.

Steps 1 and 8 are only applied to rsds and reflect that this climate variable has a physical upper bound which varies over the annual cycle. In order to fit this case into the unified framework, which at its core assumes constant bounds and thresholds, rsds values are scaled to the interval $[0,1]$ in step 1, and back to their actual range in step 8. These scalings are done using annual cycles of upper bounds that are estimated from the rsds values in $x_{\mathrm{hist}}^{\mathrm{obs}}$, $x_{\mathrm{hist}}^{\mathrm{sim}}$, $x_{\mathrm{fut}}^{\mathrm{sim}}$. Following Lange (2018), annual cycles of upper bounds at daily temporal resolution are estimated as running mean values of running maximum values of multi-year daily maximum values. Here, a window length of 31 days is used for the running window calculations. Let $b_{\mathrm{hist}}^{\mathrm{obs}}$, $b_{\mathrm{hist}}^{\mathrm{sim}}$, $b_{\mathrm{fut}}^{\mathrm{sim}}$ be these annual cycles estimated for time series $x_{\mathrm{hist}}^{\mathrm{obs}}$, $x_{\mathrm{hist}}^{\mathrm{sim}}$, $x_{\mathrm{fut}}^{\mathrm{sim}}$, respectively. Let further $x_{ij}$ be the value of one of these time series on day $j$ of year $i$, and let $b_j$ be the upper bound for that day of the year according to the corresponding annual cycle, then $x_{ij} \leq b_j$ holds true for all years $i$ and $j = 1, \ldots, 366$. The scaling in step 1 is done according to $x_{ij} \mapsto x_{ij}/b_j$. The scaling in step 8 requires an annual cycle of upper bounds to the bias-adjusted rsds values. Let $b_{\mathrm{fut}}^{\mathrm{obs}}$ denote this annual cycle. Following Frieler et al. (2017, Eq. (2)), it is estimated according to $b_{\mathrm{fut}}^{\mathrm{obs}} = b_{\mathrm{hist}}^{\mathrm{obs}} b_{\mathrm{fut}}^{\mathrm{sim}} / b_{\mathrm{hist}}^{\mathrm{sim}}$. The scaling in step 8 is then done according to $y_{ij} \mapsto y_{ij} b_j$, where $y_{ij}$ is the value of $y_{\mathrm{fut}}^{\mathrm{sim}}$ on day $j$ of year $i$, and $b_j$ is the upper bound for that day of the year according to $b_{\mathrm{fut}}^{\mathrm{obs}}$.

Step 2 is only applied to prsnratio and reflects that values of this variable are missing on days of zero precipitation, because on these days the ratio prsn/pr is not defined. In order to fit this case into the unified framework, which at its core assumes gap-less time series, missing prsnratio values are replaced by random sampling from available values. More precisely, for every missing value in $x_{\mathrm{hist}}^{\mathrm{obs}}$, $x_{\mathrm{hist}}^{\mathrm{sim}}$, $x_{\mathrm{fut}}^{\mathrm{sim}}$, an independent random number $p$ is drawn from the interval $[0,100] \subset \mathbb{R}$ with uniform probability, and the missing value is replaced by the $p$th empirical percentile of all available values in $x_{\mathrm{hist}}^{\mathrm{obs}}$, $x_{\mathrm{hist}}^{\mathrm{sim}}$, $x_{\mathrm{fut}}^{\mathrm{sim}}$, respectively. This procedure approximately preserves the distribution of values in the time series.

Steps 3 and 7 are only applied to psl, rlds, and tas, and reflect that these variables can have significant trends not only between but also within training period and application period. In order to prevent a confusion of these trends with interannual variability during quantile mapping (steps 5 and 6), linear trends within $x_{\mathrm{hist}}^{\mathrm{obs}}$, $x_{\mathrm{hist}}^{\mathrm{sim}}$, $x_{\mathrm{fut}}^{\mathrm{sim}}$ are removed in step 3 and restored in step 7. Trend lines $t_{\mathrm{hist}}^{\mathrm{obs}}$, $t_{\mathrm{hist}}^{\mathrm{sim}}$, $t_{\mathrm{fut}}^{\mathrm{sim}}$ are estimated at annual temporal resolution, i.e. by linear regression of annual mean values of the daily values of the respective time series. Let $x_{ij}$ be the value of one of these time series on day $j$ of year $i$, and let $t_i$ be the value for year $i$ of the corresponding trend line, which is shifted such that $\sum_i t_i = 0$. Then detrending in step 3 is done according to $x_{ij} \mapsto x_{ij} - t_i$, and the trend simulated within the application period is restored in step 7 according to $y_{ij} \mapsto y_{ij} + t_i$, where $y_{ij}$ is the value of $y_{\mathrm{fut}}^{\mathrm{sim}}$ on day $j$ of year $i$, and $t_i$ is the value for that year of trend line $t_{\mathrm{fut}}^{\mathrm{sim}}$.

Step 4 is only applied to bounded variables, i.e. variables which have either a lower bound (and threshold) or an upper bound (and threshold) or both (see Table 2). These bounds reflect physical limits to values these variables can take. Thresholds located slightly above the lower bound and slightly below the upper bound are used in step 6 to bias-adjust the frequencies of occurrence of values close to the bounds. In particular, the lower threshold of pr is used to bias-adjust the dry day frequency, i.e. the frequency of occurrence of $\mathrm{pr} < 0.1$ mm/day. In most cases, the simulated dry day frequency will be lower than the observed





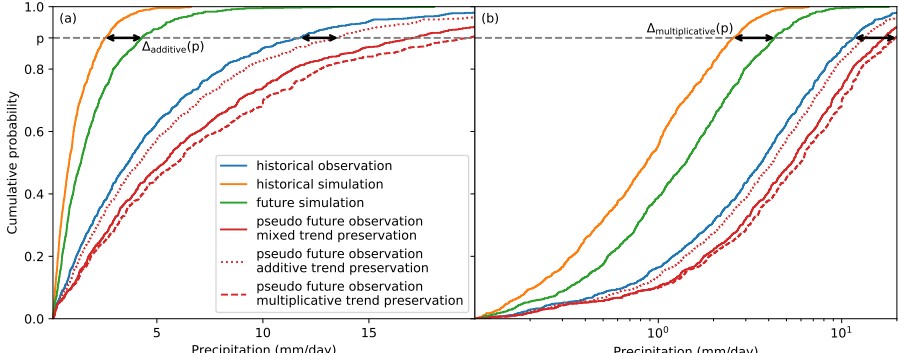

**Figure 1.** Schematic of climate change signal transfer from simulations to observations for wet day precipitation. Empirical cumulative distribution functions of historical and future simulations and observations are displayed using a linear precipitation scale in (a) and a logarithmic precipitation scale in (b). Pseudo future observations generated preserving different kinds of trends are shown in red with different line styles. For the 90th percentile, black double-headed arrows indicate additive trend preservation in (a) and multiplicative trend preservation in (b). Mixed trend preservation is explained in the text.

one (drizzle effect), and its bias adjustment is easily done by setting pr values on some initially wet days to 0. Conversely, if the simulated dry day frequency is too high, its bias adjustment requires turning initially dry days into wet days. These days are randomly selected following Cannon et al. (2015): All values below the lower threshold, $\alpha$, and above the upper threshold, $\beta$, are replaced by random numbers drawn from the open interval $(a, \alpha)$ and $(\beta, b)$, where $a$ and $b$ are the lower bound and

the upper bound, respectively. These new values can then be moved across the respective threshold by quantile mapping in step 6. In contrast to Cannon et al. (2015), random numbers from $(a, \alpha)$ and $(\beta, b)$ are not drawn with uniform probability but with power-law probability that increases towards the respective bound, as this approach is found to alleviate kinks in the distribution of wet day precipitation after bias adjustment.

    Step 5 generates pseudo future observations, which are needed for parametric quantile mapping in step 6. These pseudo

future observations are generated such that trends in all quantiles between any two application periods are approximately the same before and after quantile mapping. This makes the bias adjustment method trend-preserving in all quantiles. Different kinds of trends are preserved for different climate variables (Table 2).

    Pseudo future observations for one specific future time period are generated by transferring simulated climate change signals between the historical and the future time period to the historical observations. This transfer is done quantile by quantile using

a non-parametric kind of quantile mapping. In the following, I will describe the transfer for additive, multiplicative, mixed, and bounded trend preservation. Figure 1 provides an illustration for the former three of these four cases.

    In what follows, let $F_{\text{hist}}^{\text{obs}}$, $F_{\text{hist}}^{\text{sim}}$, $F_{\text{fut}}^{\text{sim}}$ be the empirical cumulative distribution function of all values in $x_{\text{hist}}^{\text{obs}}$, $x_{\text{hist}}^{\text{sim}}$, $x_{\text{fut}}^{\text{sim}}$, respectively. Let $Q_{\text{hist}}^{\text{obs}}$, $Q_{\text{hist}}^{\text{sim}}$, $Q_{\text{fut}}^{\text{sim}}$ be the corresponding quantile functions. Let $x$ be a value of $x_{\text{hist}}^{\text{obs}}$, let $p = F_{\text{hist}}^{\text{obs}}(x)$ be the cumulative probability of $x$, and let $y$ be the pseudo future observation corresponding to $x$. Additive trend preservation is




achieved by an additive climate change signal transfer, i.e. in this case, $y$ is generated according to

$$y = x + \Delta_{\text{additive}}(p), \text{ where} \tag{1}$$

$$\Delta_{\text{additive}}(p) = Q_{\text{fut}}^{\text{sim}}(p) - Q_{\text{hist}}^{\text{sim}}(p). \tag{2}$$

Additive trend preservation is the goal here for climate variables psl, rlds, and tas.

Multiplicative trend preservation is achieved by a multiplicative climate change signal transfer, i.e. in this case, $y$ is generated according to

$$y = x\,\Delta_{\text{multiplicative}}(p), \text{ where} \tag{3}$$

$$\Delta_{\text{multiplicative}}(p) = \max\left(0.01, \min\left(100, \Delta_{\text{multiplicative}}^{*}(p)\right)\right), \text{ and} \tag{4}$$

$$\Delta_{\text{multiplicative}}^{*}(p) = \begin{cases} 1 & \text{if } Q_{\text{hist}}^{\text{sim}}(p) = 0, \\ Q_{\text{fut}}^{\text{sim}}(p)/Q_{\text{hist}}^{\text{sim}}(p) & \text{otherwise.} \end{cases} \tag{5}$$

Note that the limits imposed in Eq. (4) are usually only reached for very small values of $x$. Multiplicative trend preservation is in most cases but not always the goal here for climate variables pr, sfcWind, and tasrange. It is *not* the goal here if $x = Q_{\text{hist}}^{\text{obs}}(p)$ is much larger than the corresponding quantile of the historical simulations $Q_{\text{hist}}^{\text{sim}}(p)$ (this corresponds to a large negative bias in the historical time period) because in this case even moderate multiplicative climate change signals $Q_{\text{fut}}^{\text{sim}}(p)/Q_{\text{hist}}^{\text{sim}}(p)$ can result in unrealistically large $y$ values, as illustrated in Fig. 1, which I want to avoid generating in particular for pr.

Pseudo future observations for pr, sfcWind, and tasrange are therefore generated by a mixed (multiplicative and additive) climate change signal transfer, i.e. for these climate variables, $y$ is generated according to

$$y = \gamma(p)\,x\,\Delta_{\text{multiplicative}}(p) + (1 - \gamma(p))(x + \Delta_{\text{additive}}(p)), \text{ where} \tag{6}$$

$$\gamma(p) = \begin{cases} 1 & \text{if } Q_{\text{hist}}^{\text{sim}}(p) \geq Q_{\text{hist}}^{\text{obs}}(p), \\ 0.5\left(1 + \cos\left(\left(Q_{\text{hist}}^{\text{obs}}(p)/Q_{\text{hist}}^{\text{sim}}(p) - 1\right)\pi/8\right)\right) & \text{if } Q_{\text{hist}}^{\text{sim}}(p) < Q_{\text{hist}}^{\text{obs}}(p) < 9\,Q_{\text{hist}}^{\text{sim}}(p), \\ 0 & \text{otherwise.} \end{cases} \tag{7}$$

This translates to a multiplicative trend preservation for positive biases, an additive trend preservation for large negative bi-
ases, and a mixed trend preservation for moderate negative biases in the historical time period. A smooth transition from multiplicative to additive trend preservation is facilitated by the function $\gamma(p)$ (Eq. (7) and Fig. 2).

For climate variables with both lower bound $a$ and upper bound $b$, climate change signals are transferred respecting these bounds, i.e. for these climate variables, $y$ is generated according to

$$y = \begin{cases} a + (x - a)\left(Q_{\text{fut}}^{\text{sim}}(p) - a\right)/\left(Q_{\text{hist}}^{\text{sim}}(p) - a\right) & \text{if } Q_{\text{hist}}^{\text{sim}}(p) > Q_{\text{fut}}^{\text{sim}}(p), \\ x & \text{if } Q_{\text{hist}}^{\text{sim}}(p) = Q_{\text{fut}}^{\text{sim}}(p), \\ b - (b - x)\left(b - Q_{\text{fut}}^{\text{sim}}(p)\right)/\left(b - Q_{\text{hist}}^{\text{sim}}(p)\right) & \text{otherwise.} \end{cases} \tag{8}$$



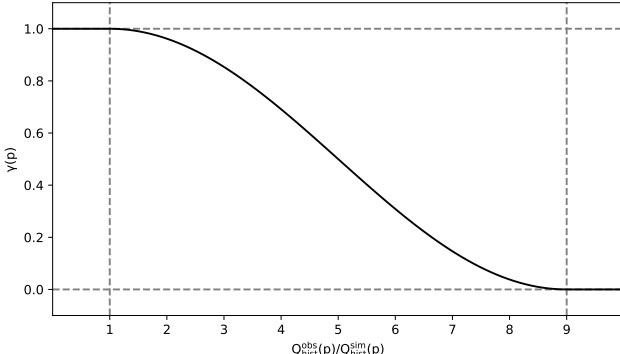

**Figure 2.** Function $\gamma(p)$ used to transition from multiplicative to additive trend preservation in mixed trend preservation (Eqs. (6–7)).

Bounded trend preservation is the goal here for climate variables hurs, prsnratio, scaled rsds, and tasskew.

Step 6 is the core of the new unified bias adjustment framework. For unbounded climate variables, it consists of a parametric quantile mapping of $x_{\mathrm{fut}}^{\mathrm{sim}}$ to the pseudo future observations generated in step 5. For climate variables with at least one bound, it consists of a bias adjustment of the frequency of values beyond threshold, and a parametric quantile mapping of all other values in $x_{\mathrm{fut}}^{\mathrm{sim}}$.

Frequencies of values beyond threshold are bias-adjusted as follows. Let $P_{\mathrm{hist}}^{\mathrm{obs}}$, $P_{\mathrm{hist}}^{\mathrm{sim}}$, $P_{\mathrm{fut}}^{\mathrm{sim}}$ be the relative frequency of values less than $\alpha$ in $x_{\mathrm{hist}}^{\mathrm{obs}}$, $x_{\mathrm{hist}}^{\mathrm{sim}}$, $x_{\mathrm{fut}}^{\mathrm{sim}}$ respectively. Similar to step 5, a pseudo future observation of this frequency, $P_{\mathrm{fut}}^{\mathrm{obs}}$, is generated by transferring the simulated climate change signal to the historically observed value,

$$
P_{\mathrm{fut}}^{\mathrm{obs}} = \begin{cases} P_{\mathrm{hist}}^{\mathrm{obs}} P_{\mathrm{fut}}^{\mathrm{sim}} / P_{\mathrm{hist}}^{\mathrm{sim}} & \text{if } P_{\mathrm{hist}}^{\mathrm{sim}} > P_{\mathrm{fut}}^{\mathrm{sim}}, \\ P_{\mathrm{hist}}^{\mathrm{obs}} & \text{if } P_{\mathrm{hist}}^{\mathrm{sim}} = P_{\mathrm{fut}}^{\mathrm{sim}}, \\ 1 - \left(1 - P_{\mathrm{hist}}^{\mathrm{obs}}\right)\left(1 - P_{\mathrm{fut}}^{\mathrm{sim}}\right) / \left(1 - P_{\mathrm{hist}}^{\mathrm{sim}}\right) & \text{otherwise.} \end{cases} \tag{9}
$$

Then, if $x_{\mathrm{fut}}^{\mathrm{sim}}$ is of length $n$, the $nP_{\mathrm{fut}}^{\mathrm{obs}}$ lowest values of $x_{\mathrm{fut}}^{\mathrm{sim}}$ are set to $a$. Similarly, for climate variables with an upper bound, the relative frequency of values greater than $\beta$ is bias-adjusted by setting the $nP_{\mathrm{fut}}^{\mathrm{obs}}$ highest values of $x_{\mathrm{fut}}^{\mathrm{sim}}$ to $b$, where $P_{\mathrm{fut}}^{\mathrm{obs}}$ is generated using Eq. (9) with relative frequencies of values less than $\alpha$ replaced by relative frequencies of values greater than $\beta$.

All other values in $x_{\mathrm{fut}}^{\mathrm{sim}}$ (or all values in the case of an unbounded climate variable) are bias-adjusted using parametric

quantile mapping, the pseudo observations $x_{\mathrm{fut}}^{\mathrm{obs}}$ generated in step 5, as well as the historical observations and simulations $x_{\mathrm{hist}}^{\mathrm{obs}}$ and $x_{\mathrm{hist}}^{\mathrm{sim}}$, respectively. Distributions used for parametric quantile mapping are the beta distribution for bounded climate variables (hurs, prsnratio, scaled rsds, tasskew), the gamma distribution for pr, the normal distribution for unbounded climate variables (psl, rlds, tas), the Weibull distribution for sfcWind, and the Rice distribution for tasrange. For unbounded climate variables, distributions are fitted to all values in $x_{\mathrm{hist}}^{\mathrm{obs}}$, $x_{\mathrm{fut}}^{\mathrm{obs}}$, $x_{\mathrm{hist}}^{\mathrm{sim}}$, $x_{\mathrm{fut}}^{\mathrm{sim}}$. For climate variables with a lower and/or upper bound,

distributions are only fitted to values greater than $\alpha$ and/or less than $\beta$ in $x_{\mathrm{hist}}^{\mathrm{obs}}$, $x_{\mathrm{fut}}^{\mathrm{obs}}$, $x_{\mathrm{hist}}^{\mathrm{sim}}$, and to all values in $x_{\mathrm{fut}}^{\mathrm{sim}}$ that where




not set to $a$ or $b$ in the first part of step 6. Let $\hat{F}_{\text{hist}}^{\text{obs}}$, $\hat{F}_{\text{fut}}^{\text{obs}}$, $\hat{F}_{\text{hist}}^{\text{sim}}$, $\hat{F}_{\text{fut}}^{\text{sim}}$ be the cumulative distribution functions of these fitted distributions.

The parametric quantile mapping method used in ISIMIP3 is inspired by the scaled distribution mapping method introduced by Switanek et al. (2017), which in addition to biases in quantiles also adjusts biases in the likelihood of individual events.

For the sake of argument, let me assume that the number of values $\hat{F}_{\text{hist}}^{\text{obs}}$, $\hat{F}_{\text{fut}}^{\text{obs}}$, $\hat{F}_{\text{hist}}^{\text{sim}}$, $\hat{F}_{\text{fut}}^{\text{sim}}$ were fitted to is the same for all four cumulative distribution functions, and let $\hat{x}_{\text{hist}}^{\text{sim}}$, $\hat{x}_{\text{hist}}^{\text{sim}}$, $\hat{x}_{\text{fut}}^{\text{sim}}$, $\hat{x}_{\text{fut}}^{\text{sim}}$ be the lowest of these values, respectively. Then $\hat{x}_{\text{fut}}^{\text{sim}}$ is quantile-mapped according to

$$\hat{x}_{\text{fut}}^{\text{sim}} \mapsto \hat{F}_{\text{fut}}^{\text{obs}\,-1}\left(\text{logit}^{-1}\left(L_{\text{hist}}^{\text{obs}} + \Delta_{\text{log-odds}}\right)\right), \text{ where} \tag{10}$$

$$\Delta_{\text{log-odds}} = \max\left(-\log 10, \min\left(\log 10, L_{\text{fut}}^{\text{sim}} - L_{\text{hist}}^{\text{sim}}\right)\right), \text{ and} \tag{11}$$

$$L_{\text{hist}}^{\text{obs}} = \text{logit}\left(\hat{F}_{\text{hist}}^{\text{obs}}\left(\hat{x}_{\text{hist}}^{\text{obs}}\right)\right), \tag{12}$$

$$L_{\text{hist}}^{\text{sim}} = \text{logit}\left(\hat{F}_{\text{hist}}^{\text{sim}}\left(\hat{x}_{\text{hist}}^{\text{sim}}\right)\right), \tag{13}$$

$$L_{\text{fut}}^{\text{sim}} = \text{logit}\left(\hat{F}_{\text{fut}}^{\text{sim}}\left(\hat{x}_{\text{fut}}^{\text{sim}}\right)\right). \tag{14}$$

Values of higher rank are quantile-mapped in the same way, i.e. using Eqs. (10–14) and $\hat{x}_{\text{hist}}^{\text{sim}}$, $\hat{x}_{\text{hist}}^{\text{sim}}$, $\hat{x}_{\text{fut}}^{\text{sim}}$, $\hat{x}_{\text{fut}}^{\text{sim}}$ of equal rank. Additional interpolations need to be introduced in Eqs. (10–14) to make them work in the general case of unequal sample sizes, as explained by Switanek et al. (2017).

Equations (10–14) result in a perfect match in distribution if training and application period are identical. In this case, $\Delta_{\text{log-odds}} = 0$ and likelihoods of events are mapped from $\hat{F}_{\text{fut}}^{\text{sim}}\left(\hat{x}_{\text{fut}}^{\text{sim}}\right) = \hat{F}_{\text{hist}}^{\text{sim}}\left(\hat{x}_{\text{hist}}^{\text{sim}}\right)$ to $\hat{F}_{\text{hist}}^{\text{obs}}\left(\hat{x}_{\text{hist}}^{\text{obs}}\right)$. In all other cases, the simulated climate change signal in event likelihood is transferred to the historically observed event likelihood such that changes in odds are multiplicatively preserved. To see that this is true, note that

$$\text{logit}\, p_1 + \text{logit}\, p_2 - \text{logit}\, p_3$$

$$= \log\left(\frac{p_1}{1-p_1}\right) + \log\left(\frac{p_2}{1-p_2}\right) - \log\left(\frac{p_3}{1-p_3}\right)$$

$$= \log\left(\frac{p_1}{1-p_1}\frac{p_2}{1-p_2}\bigg/\frac{p_3}{1-p_3}\right). \tag{15}$$

Asymptotically, i.e. for extreme values, the odds scaling used here is equivalent to the return interval scaling used by Switanek et al. (2017). The limits imposed in Eq. (11) are to prevent the generation of unrealistic event likelihoods.

Note that in contrast to all other climate variables, the likelihood of individual events is not adjusted for tas. Instead, in this case, Eqs. (10–14) are replaced by $\hat{x}_{\text{fut}}^{\text{sim}} \mapsto \hat{F}_{\text{fut}}^{\text{obs}\,-1}\left(\hat{F}_{\text{fut}}^{\text{sim}}\left(\hat{x}_{\text{fut}}^{\text{sim}}\right)\right)$. The reason for this exception is that the event likelihood adjustment can produce artifacts if large nonlinear trends are present within the training or application period. Examples of such cases have (only) been found for tas.





### 3.2.2 ISIMIP3 statistical downscaling method

As described in the introduction, the ISIMIP3 bias adjustment method shall be applied at the spatial resolution of the climate simulation data using spatially aggregated climate observation data. Since the resulting data can be considered bias-free, their subsequent statistical downscaling should be done using a method which preserves values at the aggregated spatial resolution.

The ISIMIP3 statistical downscaling method has this property. Since the new method is based on the MBCn algorithm by Cannon (2017) it is abbreviated to MBCnSD in the following.

The MBCnSD algorithm is independently applied to every climate variable and calendar month. It requires that the coarse grid of the climate simulation data and the fine grid of the climate observation data are compatible in the sense that every fine grid cell is entirely contained in one coarse grid cell. For example, that is the case if the coarse and fine grid are the global

$2° \times 2°$ and $1° \times 1°$ latitude–longitude grid, respectively, since then every coarse grid cell contains exactly $K = 4$ fine grid cells.

The MBCn algorithm by Cannon (2017) is a multivariate quantile mapping bias adjustment method. It is employed here in the context of statistical downscaling because the downscaling problem at hand can be considered as yet another bias adjustment problem: Once the climate data to be downscaled have been broadcasted to the fine grid, their statistical downscaling can be

achieved by an adjustment of the multivariate distribution of all time series contained in one coarse grid cell.

The MBCn algorithm applies a series of univariate non-parametric quantile mappings along randomly chosen axes. Mathematically, this is achieved by repeatedly rotating the climate simulation and observation data using random $K \times K$ orthogonal matrices, each time followed by $K$ univariate quantile mappings. This use of random rotation matrices makes the ISIMIP3 statistical downscaling method stochastic.

The MBCn algorithm cannot be used as is to solve the downscaling problem at hand because it does not have the required preservation property. The preservation of values at the aggregated spatial resolution translates to a preservation of the weighted sum of all time series contained in one coarse grid cell. With MBCnSD, this is achieved by an additional conservation step following the $K$ univariate quantile mappings in every iteration of the algorithm. For $K = 2$, this is illustrated in Fig. 3. A corner case of what can happen without this additional step is shown in Fig. 4: For certain axes rotation sequences, the MBCn

algorithm almost reverses the ranks of values along one axis, which results in strongly changed aggregated values. Figure 4 also exemplifies that this is prevented by the MBCnSD algorithm.

If the resolution gap between climate simulation and observation data is large then statistical downscaling can be done in one big step or in multiple small steps. Vandal et al. (2018) have shown that statistical downscaling with neural networks works better in multiple small steps. For statistical downscaling with the MBCnSD algorithm, both approaches yield similar results.

But as in the neural network case, downscaling in multiple small steps yields slightly smoother fields than downscaling in one big step (Fig. 5) and is therefore deemed the preferred approach.

In the following, I will describe the MBCnSD algorithm in detail. In this context, let $x_{ij}^{\mathrm{sim}}$ be the previously bias-adjusted climate simulation data to be statistically downscaled, with $i$ being the time index and $j$ being the coarse grid cell index. Let further $x_{ijk}^{\mathrm{obs}}$ be the historical climate observation data on the fine grid, with $i, j$ as for $x_{ij}^{\mathrm{sim}}$, and $k = 1, \ldots, K$ being the index





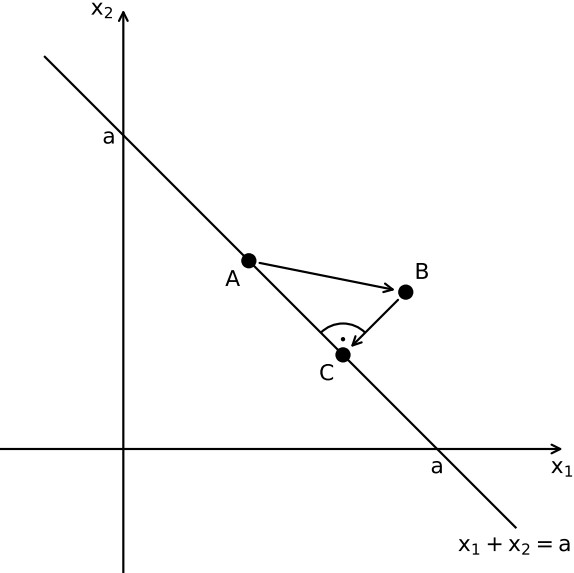

**Figure 3.** Two-dimensional illustration of one iteration of the modified MBCn algorithm used for statistical downscaling in ISIMIP3 (MBCnSD), which at its core consists of two steps. In the first step, data point $A$ is quantile-mapped to data point $B$ like in the original MBCn algorithm. In the second step, MBCnSD projects data point $B$ onto the weighted sum-preserving hyper plane of data point $A$, here with equal weights on all axes. The result is data point $C$.

for the fine grid cells contained in one coarse grid cell. Finally, let $w_{jk}$ be proportional to the area of fine grid cell $k$ in coarse grid cell $j$. Then the MBCnSD algorithm works as follows.

1. (For all variables.) Bilinearly interpolate $x_{ij}^{\mathrm{sim}}$ to the fine grid. Let $x_{ijk}^{\mathrm{sim}}$ be the result.

2. (For bounded variables only.) Randomize values beyond threshold in $x_{ij}^{\mathrm{sim}}, x_{ijk}^{\mathrm{sim}}, x_{ijk}^{\mathrm{obs}}$.

5 3. (For all variables.) Apply the core of the MBCnSD algorithm independently to every coarse grid cell $j$. Let $y_{ijk}^{\mathrm{sim}}$ be the result.

4. (For bounded variables only.) De-randomize values beyond threshold in $y_{ijk}^{\mathrm{sim}}$.

Step 1 broadcasts the previously bias-adjusted climate simulation data to the fine grid. This is done using bilinear and not conservative interpolation, which in this case would be equivalent to setting $x_{ijk}^{\mathrm{sim}} = x_{ij}^{\mathrm{sim}}$ for all $k$, because the former 10 approach results in smoother fields than the latter, as exemplified in Fig. 6. There are two reasons for that. First, bilinear interpolation already generates some of the spatial variability within each coarse grid cell that statistical downscaling has to add whereas conservative interpolation does not. Therefore, the MBCnSD algorithm would have to add more variability after conservative than bilinear interpolation, with the result of more noisy fields. Secondly, bilinear interpolation transfers spatial



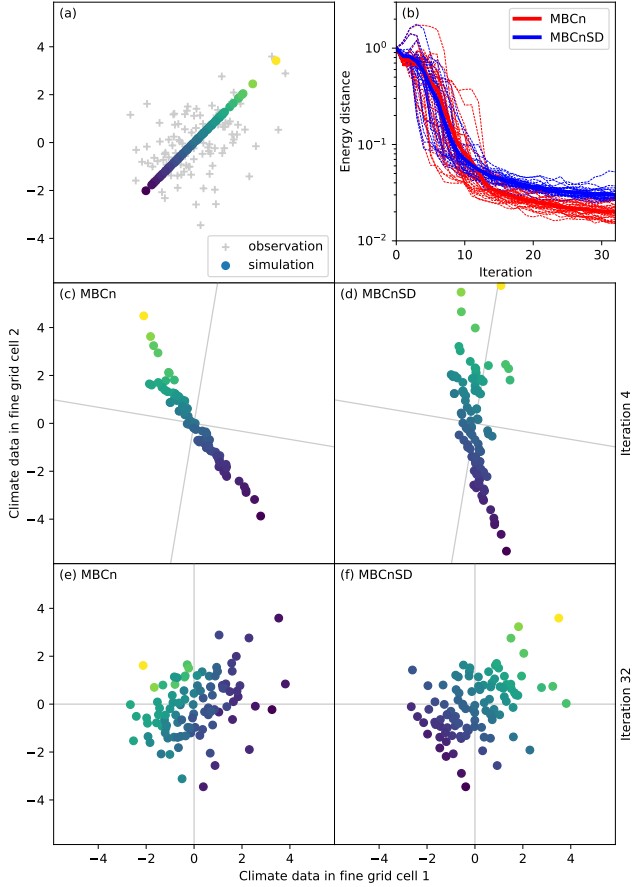

**Figure 4.** Statistical downscaling of artificial two-dimensional climate data with the original MBCn algorithm by Cannon (2017) and the modified MBCn algorithm used for statistical downscaling in ISIMIP3 (MBCnSD). Panel (a) shows an example of observation and simulation data drawn from a bivariate standard normal distribution with cross-correlation 0 and 1, respectively. Panels (c–j) show the result of statistical downscaling after 4 (c,d) and 32 (e,f) iterations of the MBCn (c,e) and MBCnSD (d,f) algorithm. To track changes, the color of a data point in panels (c-f) is the same as the color of the corresponding original data point in panel (a). Gray lines in panels (c–f) represent the axes along which univariate quantile mappings are applied in the respective iteration. Note that MBCn and MBCnSD use the same sequence of axes rotations here. Panel (b) shows the energy distance (Székely and Rizzo, 2013) between observation and simulation data over iterations for 30 random data samples (thin dashed lines) and on average over these 30 samples (thick solid lines).

gradients between coarse grid cells to the fine grid whereas conservative interpolation does not. The MBCnsD algorithm can then preserve these gradients to the degree that they are meaningful, which results in smoother fields.

Steps 2 and 4 are only applied to variables which have either a lower bound (and threshold) or an upper bound (and threshold) or both. I use the same bounds and thresholds for statistical downscaling as for bias adjustment (Table 2) for all climate variables except rsds. For statistical downscaling, rsds values are not scaled, have a lower bound of 0, a lower threshold of 0.01 W/m$^2$, and no upper bound (or threshold). The randomization itself works exactly as in step 4 of the ISIMIP3 bias adjustment method.



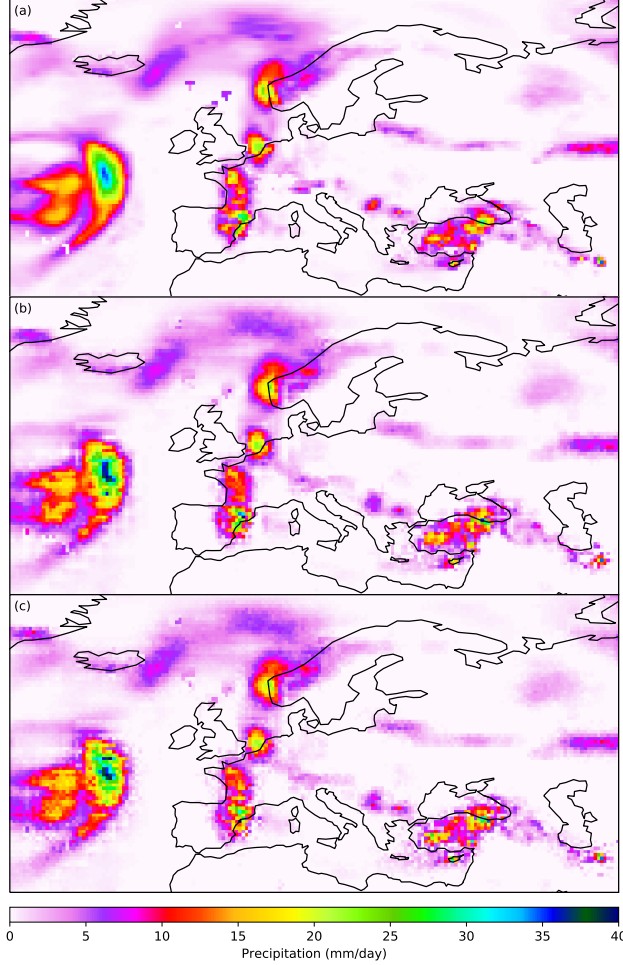

**Figure 5.** Statistical downscaling of precipitation from $2°$ to $1/2°$ spatial resolution in two small steps ($2°$ to $1°$ and $1°$ to $1/2°$) versus in one big step. In this example, the MBCnSD algorithm is applied for statistical downscaling of spatially aggregated historical observation data. Shown are precipitation fields over Europe for one particular day: The original precipitation field in panel (a) and the result of statistical downscaling in two small steps and in one big step in panel (b) and (c), respectively.

For de-randomization, all values below the lower threshold are set to the lower bound, and all values above the upper threshold are set to the upper bound. Note that the MBCnSD algorithm is applied for statistical downscaling of hurs, pr, prsnratio, psl, rlds, rsds, sfcWind, tas, tasrange, and tasskew. Bias-adjusted and statistically downscaled prsn, tasmax, and tasmin values are then derived as described in Sect. 3.2.1.

5    Step 3 is the core of the MBCnSD algorithm and is applied independently to every coarse grid cell $j$. Therefore, in the following, let $j$ be arbitrary but fixed. Let $\boldsymbol{X} = (X_i)$ be the vector with components $X_i = x_{ij}^{\mathrm{sim}}$, and let $\mathbf{Y} = (Y_{ik})$ and $\mathbf{Z} =$





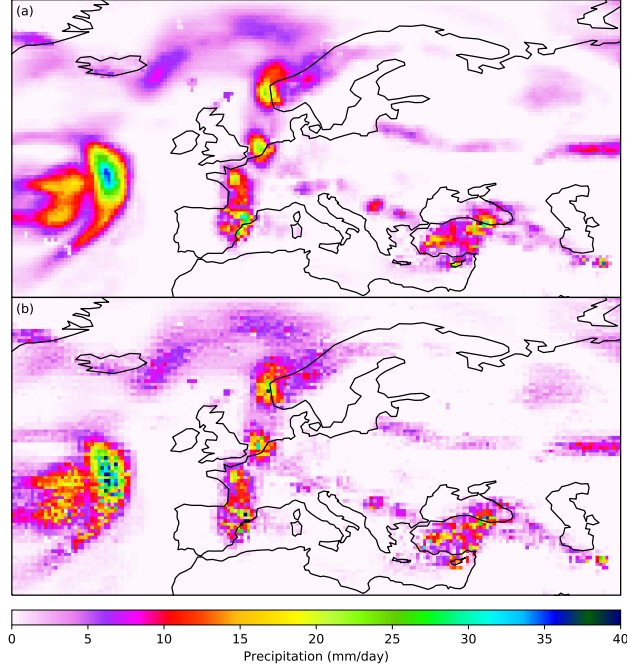

**Figure 6.** Statistical downscaling of precipitation from $1°$ to $1/2°$ spatial resolution using different broadcasting methods in step 1 of the MBCnSD algorithm. In this example, the algorithm is applied for statistical downscaling of spatially aggregated historical observation data. Shown are downscaled precipitation fields over Europe for one particular day, using bilinear and conservative interpolation for broadcasting in panel (a) and (b), respectively. The corresponding original precipitation field is shown in panel (a) of Fig. 5.

$(Z_{ik})$ be matrices with components $Y_{ik} = x^{\text{sim}}_{ijk}$ and $Z_{ik} = x^{\text{obs}}_{ijk}$, respectively. Let further $w = \sum_k w_{jk}$ be the sum of all fine grid cell area weights and let $\tilde{w} = \sqrt{\sum_k w^2_{jk}}$ be their root sum square. Let $\boldsymbol{W} = (W_k)$ be the vector with components $W_k = w_{jk}/\tilde{w}$.

The core of the MBCnSD algorithm proceeds in three sub-steps, which I will refer to as 3a, 3b, and 3c in the following. Sub-step 3a adjusts $\mathbf{Y}$ to the end of restoring the spatially aggregated values which shall be preserved by the algorithm but

5 have been altered by bilinear interpolation in step 1. In addition, sub-step 3a adjusts $\mathbf{Z}$ to the end of transferring the simulated climate change signal to the historical climate observation data on the fine grid. This signal is incorporated in $\boldsymbol{X}$, given that $\boldsymbol{X}$ is the result of quantile mapping to the pseudo future climate observation data generated in step 5 of the ISIMIP3 bias adjustment algorithm. Since the simulated climate change signal is only available at the coarse resolution, it is transferred to $\mathbf{Z}$ at that resolution, and all statistical dependencies at higher resolution are left unchanged. Mathematically, sub-step 3a proceeds

10 as follows.

1. Generate a $K \times K$ orthogonal matrix $\mathbf{O}$ whose first column is equal to $\boldsymbol{W}$. Set $\mathbf{O}_{\text{total}} = \mathbf{O}$. Rotate $\mathbf{Y}, \mathbf{Z}, \boldsymbol{W}$ using $\mathbf{O}$.

2. Set $Y_{i1}$ to $X_i w/\tilde{w}$ to restore the spatially aggregated values.

3. Do a non-parametric quantile mapping of $(Z_{i1})$ to $(X_i w/\tilde{w})$ to transfer the simulated climate change signal.





Here and in the following, to rotate $\mathbf{Y}, \mathbf{Z}, \boldsymbol{W}$ using $\mathbf{O}$ means to apply the matrix multiplications

$$\mathbf{Y} \mapsto \mathbf{Y}\mathbf{O}, \tag{16}$$

$$\mathbf{Z} \mapsto \mathbf{Z}\mathbf{O}, \tag{17}$$

$$\boldsymbol{W} \mapsto (\boldsymbol{W}^T \mathbf{O})^T = \mathbf{O}^T \boldsymbol{W}, \tag{18}$$

and to set $\mathbf{Y}, \mathbf{Z}, \boldsymbol{W}$ to the respective result. To do a non-parametric quantile mapping of $(A_i)$ to $(B_i)$ means to use empirically estimated quantiles $a_p$ and $b_p$ of $(A_i)$ and $(B_i)$, respectively, corresponding to cumulative probabilities $p \in \{0\%, 2\%, 4\%, \ldots, 100\%\}$, with $a_0 = \min_i A_i$, $a_1 = \max_i A_i$, and the same for $b_0$, $b_1$, to define a transfer function $f$ using the linearly interpolated quantile–quantile pairs $(a_p, b_p)$, to then map $A_i \mapsto f(A_i)$, and to set $A_i$ to the result for all $i$.

In sub-step 3b, the following three steps are repeated either a fixed number of times or until $\mathbf{Y}$ has converged to $\mathbf{Z}$ in
distribution. The last two of these steps are illustrated in Fig. 3.

1. Generate a random $K \times K$ orthogonal matrix $\mathbf{O}$. Rotate $\mathbf{O}_{\text{total}}, \mathbf{Y}, \mathbf{Z}, \boldsymbol{W}$ using $\mathbf{O}$. Set $\tilde{\mathbf{Y}} = \mathbf{Y}$.

2. For all $k$, do a non-parametric quantile mapping of $(Y_{ik})$ to $(Z_{ik})$.

3. Project $\mathbf{Y}$ onto the weighted sum-preserving hyper plane of $\tilde{\mathbf{Y}}$ by subtracting $((\mathbf{Y} - \tilde{\mathbf{Y}})\boldsymbol{W}) \otimes \boldsymbol{W}$ from $\mathbf{Y}$.

In the first of these steps, random orthogonal matrices are drawn from the circular real random matrix ensemble using the
algorithm by Mezzadri (2007). In the last step, $\otimes$ denotes the outer product of two vectors. Note that the results presented in Sect. 4 are obtained using a fixed number of 20 iterations in sub-step 3b of the MBCnSD algorithm, as this was deemed sufficient for the MBCn algorithm by Cannon (2017).

In sub-step 3c, all data are rotated back to the original axes. A last quantile mapping along these axes ensures that there are no values out of bounds in the resulting data. Mathematically, sub-step 3c proceeds as follows.

1. Set $\mathbf{O} = \mathbf{O}_{\text{total}}^T$. Rotate $\mathbf{Y}, \mathbf{Z}, \boldsymbol{W}$ using $\mathbf{O}$.

2. For all $k$, do a non-parametric quantile mapping of $(Y_{ik})$ to $(Z_{ik})$.

For the arbitrary but fixed coarse grid cell $j$, the result $y_{ijk}^{\text{sim}}$ of step 3 of the MBCnSD algorithm is then given by $y_{ijk}^{\text{sim}} = Y_{ik}$.

### 3.3   Comparison

In the following section I will compare results obtained with the ISIMIP2b and ISIMIP3 bias adjustment and statistical down-
scaling methods. I will begin with results of bias adjustment applied at $2°$ spatial resolution, i.e. using climate simulation and observation data both on the global $2° \times 2°$ latitude–longitude grid. In particular, I will compare the methods' ability to (i) adjust biases in percentiles of distributions of daily values and (ii) preserve trends in these percentiles. Percentiles chosen for this comparison are the 5th, 50th, and 95th, representing the lower tail, the center, and the upper tail of a distribution, respectively. An exception is made for pr and prsn, for which instead of the 5th percentile I consider the dry day frequency, i.e. the frequency



of precipitation or snowfall flux to be less than 0.1 mm/day, and instead of the 50th and 95th percentile of all values I consider the 50th and 95th percentile of all values which exceed 0.1 mm/day, i.e. the 50th and 95th percentile of wet day precipitation.

I will then compare results obtained with the ISIMIP2b and ISIMIP3 statistical downscaling methods. To that end, both downscaling methods are combined with the ISIMIP3 bias adjustment method. These bias adjustment–statistical downscaling

method combinations are abbreviated with LI+BA and BA+SD in the following. The combination LI+BA is to represent the ISIMIP2b approach to statistical downscaling and therefore consists of a bilinear interpolation of the climate simulation data from their original spatial resolution of $2°$ to $1/2°$ followed by a bias adjustment using the climate observation data at their original spatial resolution of $1/2°$. The combination BA+SD is to represent the ISIMIP3 approach to statistical downscaling and therefore consists of a bias adjustment at $2°$ spatial resolution using the climate observation data aggregated to that

resolution followed by statistical downscaling from $2°$ to $1/2°$ in two steps ($2°$ to $1°$ and $1°$ to $1/2°$) using the ISIMIP3 statistical downscaling method.

Results at $2°$ and $1/2°$ spatial resolution will first be assessed based on the same metrics used to compare the ISIMIP2b and ISIMIP3 bias adjustment methods at $2°$ spatial resolution. This is done to demonstrate that the BA+SD approach does not impair data quality with regard to bias adjustment or trend preservation relative to the LI+BA approach. For the comparison at

$2°$ spatial resolution, the bias-adjusted and statistically downscaled climate simulation data are conservatively aggregated back to that resolution.

Secondly, I will compare results at $1/2°$ spatial resolution with regard to spatial variability within $2° \times 2°$ grid cells. Two ways of placing these grid cells will be considered. The first way is to place their centers at odd-numbered latitudes and longitudes (measured in degrees). These grid cells constitute the regular $2° \times 2°$ latitude–longitude grid of the original climate

simulation data. The second way is to place their centers at even-numbered latitudes and longitudes (measured in degrees). These grid cells form a grid that is staggered by $1°$ latitude and $1°$ longitude relative to the regular one.

For $2° \times 2°$ grid cells placed in the first way it is expected that spatial variability within them is better adjusted by BA+SD than by LI+BA by design. It is less clear if this also holds true for spatial variability within staggered $2° \times 2°$ grid cells since these contain time series whose statistical dependence is not adjusted by the ISIMIP3 statistical downscaling method. In both

cases, spatial variability within a $2° \times 2°$ grid cell is measured by the root-mean-square deviation (RMSD) of the 16 time series contained in that grid cell from their spatial average: Let $x_{ij}$ be the value on day $i$ in $1/2° \times 1/2°$ grid cell $j$. Then, for time series of length $n$, the RMSD is calculated according to

$$\text{RMSD} = \sqrt{\frac{1}{n}\sum_{i=1}^{n}\frac{1}{16}\sum_{j=1}^{16}\left(x_{ij} - \frac{1}{16}\sum_{k=1}^{16}x_{ik}\right)^2}. \tag{19}$$

The comparison of methods with regard to their ability to adjust biases and spatial variability is done in a cross-validation

framework. This is done to prevent different extents of overfitting by different methods to dominate differences in results. I first use odd-numbered years from the time period 1980–2015 for training and even-numbered years from the same time period for application. Secondly, I swap these training and application years. Finally, I merge the results of application to odd-numbered and even-numbered years to arrive at bias-adjusted and statistically downscaled data for cross-validation which fully cover the



1980–2015 time period. For the comparison of methods with regard to trend preservation, I use the full 1980–2015 time period for training and the full time periods 1980–2015 and 2064–2099 for application.

The metrics introduced above (RMSD, dry day frequency, percentiles) are calculated independently for every data set (climate observations, original climate simulations, climate simulations bias-adjusted with the ISIMIP2b/ISIMIP3 method, climate simulations bias-adjusted and statistically downscaled with the LI+BA/BA+SD method combination), climate variable, calendar month, and grid cell. The goodness of spatial variability adjustment, trend preservation, and bias adjustment is then quantified using absolute errors: For a fixed metric, adjustment method, climate variable, calendar month, and grid cell, let $x_{\mathrm{hist}}^{\mathrm{obs}}, x_{\mathrm{hist}}^{\mathrm{sim}}, x_{\mathrm{fut}}^{\mathrm{sim}}, y_{\mathrm{hist}}^{\mathrm{sim}}$, and $y_{\mathrm{fut}}^{\mathrm{sim}}$ represent values of the metric calculated for historical observations, historical simulations, future simulations, adjusted historical simulations, and adjusted future simulations, respectively. Then, in the case of spatial variability adjustment and bias adjustment, the absolute error $e$ is calculated according to

$$e = \left| y_{\mathrm{hist}}^{\mathrm{sim}} - x_{\mathrm{hist}}^{\mathrm{obs}} \right|. \tag{20}$$

In the case of trend preservation, the absolute error $e$ is calculated according to

$$e = \left| \left( y_{\mathrm{fut}}^{\mathrm{sim}} - y_{\mathrm{hist}}^{\mathrm{sim}} \right) - \left( x_{\mathrm{fut}}^{\mathrm{sim}} - x_{\mathrm{hist}}^{\mathrm{sim}} \right) \right|. \tag{21}$$

Values of these errors are then aggregated over all calendar months and grid cells using the grid cell area-weighted median. For prsn I only aggregate errors from higher than $60^{\circ}$ latitude. The aggregated values are then used to comparatively assess method performance.

## 4 Results

In the following I will first present results obtained with the ISIMIP2b and ISIMIP3 bias adjustment methods applied at $2^{\circ}$ spatial resolution. I will then compare results obtained with the ISIMIP2b and ISIMIP3 statistical downscaling methods applied for downscaling from $2^{\circ}$ to $1/2^{\circ}$ spatial resolution.

### 4.1 Comparison of bias adjustment methods

The goodness of bias adjustment and trend preservation by the ISIMIP2b and ISIMIP3 bias adjustment methods is assessed based on Fig. 7, which shows how well these methods adjust biases and preserve trends in the 5th, 50th, and 95th percentile of daily values of the ten climate variables listed in Table 1 (mind the special treatment of pr and prsn described in Sect. 3.3). Results suggest that in most calendar months and grid cells, biases are better adjusted by the ISIMIP3 method than by the ISIMIP2b method for all ten climate variables. The greatest gains are found for hurs, rlds, rsds, and sfcWind. The least yet still considerable gains are found for tas, tasmax, and tasmin. Intermediate gains are found for pr, prsn, and psl.

Results further suggest that in most calendar months and grid cells, trends in psl, rlds, and tas are considerably better preserved by the ISIMIP3 method than by the ISIMIP2b method. Trends in hurs, rsds, sfcWind, tasmax, and tasmin are mostly better preserved by the ISIMIP3 method than by the ISIMIP2b method, yet there are a few exceptions of slightly better





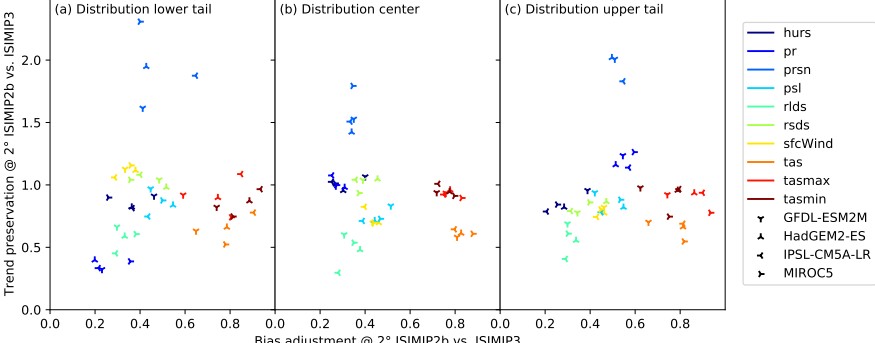

**Figure 7.** Goodness of bias adjustment (x-axis) and trend preservation (y-axis) by the ISIMIP2b and ISIMIP3 bias adjustment methods for the lower tail (a), center (b), and upper tail (c) of the distribution of daily values of ten different climate variables (color, Table 1) simulated by 4 different climate models (symbols) at $2°$ spatial resolution. Values on both axes represent ratios of spatiotemporally aggregated absolute errors after bias adjustment with the two methods (see Sect. 3.3). Values greater than 1 indicate better bias adjustment or trend preservation by the ISIMIP2b method than by the ISIMIP3 method, and vice versa for values less than 1.

trend preservation by the ISIMIP2b method for these climate variables. For pr, results suggest that the ISIMIP3 method is much better at preserving trends in dry day frequency, while both methods are similarly good at preserving trends in the 50th percentile of wet day precipitation, and the ISIMIP2b method is a bit better at preserving trends in the 95th percentile of wet day precipitation. Trends in prsn are generally better preserved by the ISIMIP2b method, presumably because the prsn/pr ratio

is left unchanged by this method whereas it is adjusted and therefore changed by the ISIMIP3 method.

### 4.2    Comparison of statistical downscaling methods

The goodness of bias adjustment and trend preservation by the LI+BA and BA+SD method combinations applied for bias adjustment and statistical downscaling from $2°$ to $1/2°$ spatial resolution is assessed based on Figs. 8 and 9, which show how well they adjust biases and preserve trends in the 5th, 50th, and 95th percentile of daily values of the ten climate variables listed

in Table 1 (mind again the special treatment of pr and prsn described in Sect. 3.3). The goodness of bias adjustment is assessed at $2°$ and $1/2°$ spatial resolution in Fig. 8 and 9, respectively, while the goodness of trend preservation is only assessed at $2°$ spatial resolution (Fig. 8) because simulated trends are only available at that resolution.

Differences between LI+BA and BA+SD in their ability to adjust biases at $2°$ spatial resolution reflect structural differences between the ISIMIP2b and ISIMIP3 approaches to statistical downscaling. Bias adjustment in BA+SD is carried out at $2°$

spatial resolution and followed by a statistical downscaling which approximately preserves values at that resolution. Therefore, biases can be expected to be well adjusted at $2°$ spatial resolution by BA+SD. In contrast, bias adjustment in LI+BA follows a bilinear interpolation to $1/2°$ spatial resolution and is independently applied to every $1/2° \times 1/2°$ grid cell. Spatial dependencies between time series within $2° \times 2°$ grid cells are not adjusted. Therefore, biases at $2°$ spatial resolution are expected to be better adjusted by BA+SD than LI+BA. Results shown in Fig. 8 are largely in line with this expectation. The greatest gains





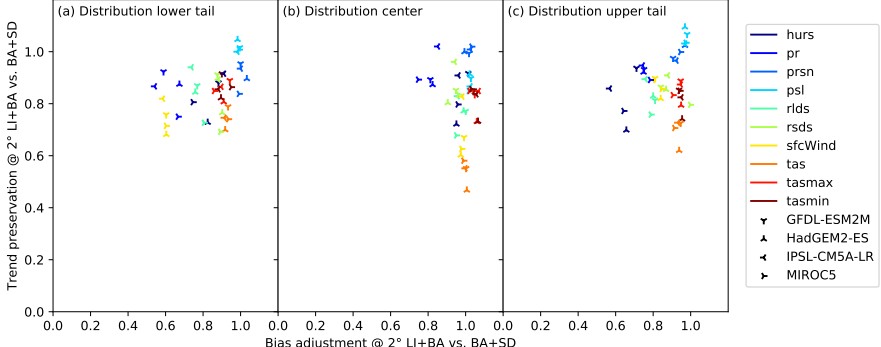

**Figure 8.** Same as Fig. 7 but for LI+BA and BA+SD applied for bias adjustment and statistical downscaling from $2°$ to $1/2°$ spatial resolution. Like in Fig. 7, absolute errors are calculated at $2°$ spatial resolution based on conservatively aggregated bias-adjusted and statistically downscaled climate simulation data.

are found for pr. Only biases in the center of the distribution of tasmax and tasmin are slightly better adjusted by LI+BA than by BA+SD in most calendar months and $2° \times 2°$ grid cells.

At $2°$ spatial resolution, BA+SD is expected to outperform LI+BA with regard to trend preservation for the same structural reason as with regard to bias adjustment. Results (Fig. 8) are in line with this expectation. Only for pr, prsn, and psl there
are cases in which LI+BA preserves trends slightly better than BA+SD. Otherwise, trends are better preserved by BA+SD, by considerable margins in particular for sfcWind and tas.

Biases at $1/2°$ spatial resolution (Fig. 9) are slightly better adjusted by BA+SD than by LI+BA in most calendar months and grid cells for all climate variables except pr and prsn. Dry day frequency biases are better adjusted by LI+BA than by BA+SD, arguably because the parametric bias adjustment of pr that is done following a bilinear interpolation in LI+BA adjusts
it explicitly and therefore precisely whereas the non-parametric quantile mapping that is used for statistical downscaling after bias adjustment in BA+SD adjusts it implicitly and only approximately (see Sect. 3.2). Biases in the 50th percentile of wet day precipitation are slightly better adjusted by LI+BA than by BA+SD in most calendar months and grid cells. The opposite is true for the 95th percentile of wet day precipitation.

In order to assess the goodness of bias adjustment across spatial scales, the $y = 1/x$ line in Fig. 9 is considered to separate
cases in which BA+SD outperforms LI+BA (below the line) from cases in which LI+BA outperforms BA+SD (above the line). Results suggest that BA+SD adjusts biases better than LI+BA in the vast majority of cases.

The goodness of spatial variability adjustment by LI+BA and BA+SD is assessed based on Fig. 10, which shows how well these method combinations adjust spatial variability within regular and staggered $2° \times 2°$ grid cells. Results suggest that, as expected, spatial variability within regular $2° \times 2°$ grid cells is better adjusted by BA+SD than by LI+BA for most calendar
months and grid cells in all cases but one (prsn simulated by HadGEM2-ES).

Spatial variability within staggered $2° \times 2°$ grid cells is better adjusted by BA+SD than by LI+BA in most cases for hurs, tas, tasmax, and tasmin, and vice versa for prsn and psl. Results are mixed for pr, rlds, rsds, and sfcWind. In order to assess how



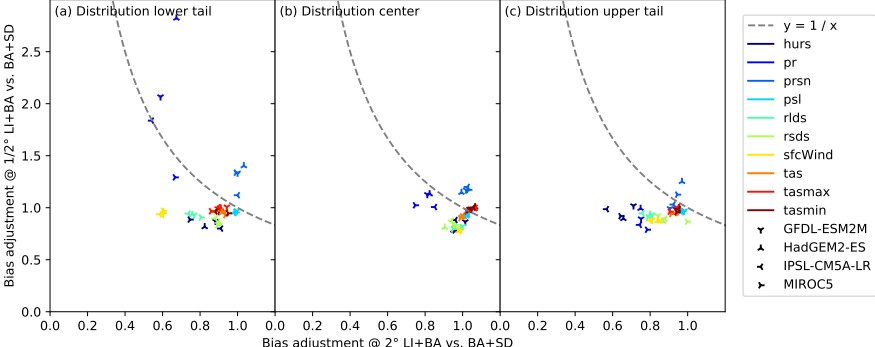

**Figure 9.** Same as Fig. 8 but with goodness of bias adjustment at $1/2°$ spatial resolution on the y-axis.

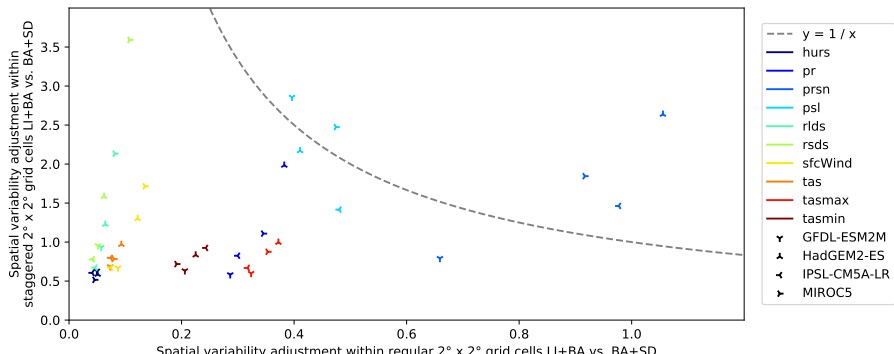

**Figure 10.** Goodness of adjustment of spatial variability within regular (x-axis) and staggered (y-axis) $2° × 2°$ grid cells by the LI+BA and BA+SD bias adjustment and statistical downscaling methods for ten different climate variables (color, Table 1) simulated by 4 different climate models (symbols). Values on both axes represent ratios of spatiotemporally aggregated absolute errors after bias adjustment and statistical downscaling with the two methods (see Sect. 3.3). Values greater than 1 indicate better spatial variability adjustment by LI+BA than by BA+SD, and vice versa for values less than 1.

spatial variability is adjusted overall, the $y = 1/x$ line in Fig. 10 is considered to separate cases in which BA+SD outperforms LI+BA (below the line) from cases in which LI+BA outperforms BA+SD (above the line). Results suggest that BA+SD adjusts spatial variability better than LI+BA in the vast majority of cases.

## 5 Conclusions

5 The ISIMIP3 bias adjustment and statistical downscaling methods outperform their predecessors in several respects. The new trend-preserving parametric quantile mapping method used for bias adjustment preserves trends and adjusts biases in distribution quantiles more accurately than the ISIMIP2b bias adjustment method. The new stochastic method used for statistical downscaling prevents the variability inflation caused by spatial interpolation in ISIMIP2b.




A major fraction of the bias adjustment gains can be attributed to the newly introduced adjustment of the likelihood of individual events. This new feature effectively corrects for the imperfections of the distribution fits that are the basis of parametric quantile mapping. In addition, it simplifies the confinement of extreme values to the physically plausible range.

Trend preservation works better with the new methods because they apply it to all distribution quantiles compared an application to only distribution mean values for most climate variables in ISIMIP2b. In addition, the new approach of bias adjustment at the spatial resolution of the climate simulation data followed by statistical downscaling to the spatial resolution of the climate observation data ensures that trends are preserved at the spatial resolution at which they were simulated.

The new approach also better adjusts spatial variability at the spatial resolution of the climate observation data than the old approach of a bilinear interpolation of climate simulation data to the spatial resolution of the climate observation data followed by bias adjustment of these interpolated data. Overall, the results presented in this paper can be considered as a proof of concept of the new paradigm of a clear separation of bias adjustment and statistical downscaling.

*Code and data availability.* The ISIMIP3 bias adjustment and statistical downscaling code is publicly available at https://doi.org/10.5281/zenodo.2549632. The ISIMIP2b bias adjustment code is publicly available at https://doi.org/10.5281/zenodo.1069050. The EWEMBI data set is publicly available via https://doi.org/10.5880/pik.2016.004. The CMIP5 multi-model ensemble output is publicly available via https://doi.org/10.1594/WDCC/CMIP5.NGEMhi for GFDL-ESM2M historical, https://doi.org/10.1594/WDCC/CMIP5.NGEMr8 for GFDL-ESM2M rcp85, https://doi.org/10.1594/WDCC/CMIP5.MOGEhi for HadGEM2-ES historical, https://doi.org/10.1594/WDCC/CMIP5.MOGEr8 for HadGEM2-ES rcp85, https://doi.org/10.1594/WDCC/CMIP5.IPILhi for IPSL-CM5A-LR historical, https://doi.org/10.1594/WDCC/CMIP5.IPILr8 for IPSL-CM5A-LR rcp85, https://doi.org/10.1594/WDCC/CMIP5.MIM5hi for MIROC5 historical, and https://doi.org/10.1594/WDCC/CMIP5.MIM5r8 for MIROC5 rcp85.

*Competing interests.* The author declares that no competing interests are present.

*Acknowledgements.* The author is grateful to Alex Cannon, Matt Switanek, Douglas Maraun, Simon Treu, Jens Heinke, and Katja Frieler for various helpful discussions at different stages of this work. He acknowledges the World Climate Research Programme's Working Group on Coupled Modelling, which is responsible for CMIP, and he thanks the climate modeling groups for producing and making available their model output. This work has received funding from the European Union's Horizon 2020 research and innovation programme under grant agreement no. 641816 Coordinated Research in Earth Systems and Climate: Experiments, kNowledge, Dissemination and Outreach (CRESCENDO).



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
