# Peer review of "Trend-preserving bias adjustment and statistical downscaling with ISIMIP3BASD (v1.0)"

_Geoscientific Model Development, 2019_

## Referee Comment (RC1) · Anonymous Referee #1 · 15 Apr 2019

This paper presents the development, implementation, and evaluation of an evolved and generalized bias correction method tailored to the ISIMIP. The method is a significant evolution from its predecessor because it separates the bias correction (BC) and statistical downscaling (SD) portions explicitly, accounts for likelihoods of individual extreme events, and recasts the BC of individual variables in one single formalism. I have carefully read the specific choices for each of the 10 corrected variables and I find them to be reasonable and in line with best practice. For example, the use of upper and lower bounds and upper and lower thresholds is a logical extension of the common dry-day correction for precipitation. Similarly, the choice to draw random numbers from a power law rather than a uniform distribution (for those variables that require it) is an intelligent improvement. The same can be said about the introduction of the Logit function. The
paper is well written, as succinct as possible and exhaustive. The illustrations are all useful and clear. I find that this work should be accepted after the author addresses a few minor issues. 1) If I understand correctly, the author uses univariate bias correction for the coarse grid. I accept that this is a progression from the former bias correction method (ISIMIP2) but, since the author does use MBC with an added conservation step (MBCnSD) for the fine grid, could this method not be applied to the course grid to begin with? Would the method not be far simpler? I would be interested in the author's reason for using univariate BC for the course grid. 2) The cross-validation is done by training on even years and validating on odd years (in a second step validation and training years are swapped). I would have divided the observational time-period in 2 consecutive and non-overlapping periods. The first for observation and the second for validation. This would allow for longer-term climate variability to be represented in the cross-validation effort. I think the author should write a few words explaining why his choice of cross-validation time periods is optimal.

Other minor issues: Page 2, line 16) The author claims to develop a new quantile mapping method that allows for controlled adjustment of biases in all quantiles. As far as I know, most QM bias correction methods do just that. Have I misunderstood something here? Page 4 line 26) "...tasmax, and tasmin are not adjusted directly. Instead, I adjust biases in pr and prsnratio..." The author passes, here and in other parts of the article, from a passive to an active (first person) voice. I suggest he chose one or the other. Page 7 line 19) x should be x\_hist^obs, am I right? Also, y needs indexes as well, I think. This should be the case in all the math that follows. I understand that the author is trying to keep the formalism light. I suggest then that the author state that, in what follows, x is actually x\_hist^obs, etc. . Caption figure 4) 3rd Sentence : Panele (c-j) should perhaps be (c-f)? Page 12 line 8/9) Write "instead of conservative" in the place of "and not conservative". Else it is not clear that we are comparing bilinear with conservative methods.

2019.

---

## Referee Comment (RC2) · Stefan Hagemann (Referee) · 16 Apr 2019

**Manuscript:** Trend-preserving bias adjustment and statistical downscaling with ISIMIP3BASD (v1.0)

**Major remarks**

The author presents a documentation about an improved bias correction technique that separates bias correction and statistical downscaling. The manuscript provides an useful overview on the bias correction method that is supposed to be used in the third phase of the Inter-Sectoral Impact Model Intercomparison Project (ISIMIP). The paper is generally written well, and the describe formalisms are necessary but not too overloaded with equations. I have only a few remarks that should be addressed.

1. Section structures should be indicated in the beginning of a section. Currently, in several cases, a description is given where information is missing. The missing information is then provided in the one of the next paragraphs, but in the beginning there was no indication that this is being done. One example for this is sect. 3.2.2., where the detailed description starts on p.11 – line 32 after 5 paragraphs of general description of the downscaling method.

2. Observations used for bias correction are usually available on a fine, regular grid while the coarse GCM grids are usually not regular (e.g. often a Gaussian grid is used). Thus, the usage of the downscaling technique must be clarified for this common case where the fine grid does not fit into the coarse grid. In this respect I was really wondering why it is written on p. 11 – line 7-11 that 'It requires that the coarse grid of the climate simulation data and the fine grid of the climate observation data are compatible …'

3. Even though the manuscript is a documentation on the technical method of the ISIMIP3 bias correction, it should not be concealed that the bias correction cannot compensate for several types of GCM errors, e.g. erroneous shifts of storm tracks in the GCM. In such cases, bias correction may even lead to erroneous climate change signals. With respect to this topic, I suggest pointing towards Maraun, D., T. Shepherd, M. Widmann, G. Zappa, D. Walton, J.M. Gutiérrez, S. Hagemann, I. Richter, P. Soares, A. Hall and L. Mearns (2017). Towards process-informed bias correction of climate change simulations. Nature Clim. Change 7: 764-773, doi: 10.1038/nclimate3418.

In summary, I suggest accepting the paper for publication after minor revisions are conducted.

**Minor remarks**

In the following suggestions for editorial corrections are marked in *Italic*.

p.3 – line 2
It is written:
'… these data are bilinearly interpolated…'

Does 'these data' refers to the original GCM data or to the data interpolated on the 2° grid?

I guess you mean the original data as otherwise the bilinear interpolation would not make so much sense as the 0.5° data fit to the 2° data.

p.3 – line 8
As the EWEMBI refers to a published dataset, but not peer-reviewed publication, I suggest including more information about the data, variables (and their respective source) and bias-correction. This may also be done in an appendix.

p.5 – Table 2
In table or the associated text, the difference between bounds and threshold should be clearly described.
In addition, it should be noted where the different types of trend preservation are prescribed (see also major remark 1).

p.9 – line 6
Is this the threshold listed in Table 2?

p.9 – line 7-13
The definitions and differences of a and $\alpha$ as wells as b and $\beta$ are unclear. Please clarify!

p.10 – eq. 10-14
I am not familiar with the logit function, and I assume other readers many not either. Please explain what this function is doing.
Or is this just the function you define in eq. 15?
This must be indicated when the logit term is used for the first time.

p.11 – line 3
It is written:
'Since the resulting data can be considered bias-free …'

The uninformed user may understand this misleading statement wrongly. See also major remark 3.

p.12 – line 8-10
Sentence "This is done …" is too long and difficult to read. Please separate into two sentences.

p.15 – line 11
The first column is set to $W$. What about the other columns?

p.17 – line 5 and 8
Replace 'is to represent' by 'represents'.

p.19 – Fig. 7 (also Fig. 8, 9, 10)
Some of the bluish and greenish colours are difficult to separate. Please improve colour setting.

---

## Author Response (AR1)

Author's response (black) to comments by the Anonymous Referee (orange)

This paper presents the development, implementation, and evaluation of an evolved and generalized bias correction method tailored to the ISIMIP. The method is a significant evolution from its predecessor because it separates the bias correction (BC) and statistical downscaling (SD) portions explicitly, accounts for likelihoods of individual extreme events, and recasts the BC of individual variables in one single formalism. I have carefully read the specific choices for each of the 10 corrected variables and I find them to be reasonable and in line with best practice. For example, the use of upper and lower bounds and upper and lower thresholds is a logical extension of the common dry-day correction for precipitation. Similarly, the choice to draw random numbers from a power law rather than a uniform distribution (for those variables that require it) is an intelligent improvement. The same can be said about the introduction of the Logit function. The paper is well written, as succinct as possible and exhaustive. The illustrations are all useful and clear. I find that this work should be accepted after the author addresses a few minor issues.

Thank you for this positive introduction.

1) If I understand correctly, the author uses univariate bias correction for the coarse grid. I accept that this is a progression from the former bias correction method (ISIMIP2) but, since the author does use MBC with an added conservation step (MBCnSD) for the fine grid, could this method not be applied to the course grid to begin with? Would the method not be far simpler? I would be interested in the author's reason for using univariate BC for the course grid.

Thank you for this suggestion. When I set out to develop ISIMIP3BASD, I actually considered making it a multivariate bias adjustment method. Then, in the rather lengthy development process, I abandoned this goal for various reasons, and later forgot about altogether. Yet I agree that ISIMIP3BASD v1.0 can easily be made multivariate. I will therefore make ISIMIP3BASD v2.0 multivariate by employing the MBCn algorithm for an adjustment of the multivariate rank distribution at the coarse grid. This adjustment will be inserted between steps 4 and 5 of the ISIMIP3BASD v1.0 bias adjustment algorithm. Otherwise, v1.0 and v2.0 of ISIMIP3BASD will be identical. I will keep using the methods described in this paper for the adjustment of the marginal distributions because these parametric quantile mapping methods promise more robust extreme value adjustments than non-parametric quantile mapping methods such as the quantile delta mapping method shipped with the original MBCn algorithm. I will add a paragraph to the Conclusions section announcing this enhancement.

As far as I can see, it is not possible to simultaneously apply MBCnSD to multiple variables such that values of all of these variables are conserved at the coarse grid scale. That is why I think that ISIMIP3BASD cannot be made much simpler.

2) The cross-validation is done by training on even years and validating on odd years (in a second step validation and training years are swapped). I would have divided the observational time-period in 2 consecutive and non-overlapping periods. The first for observation and the second for validation. This would allow for longer-term climate variability to be represented in the cross-validation effort. I think the author should write a few words explaining why his choice of cross-validation time periods is optimal.

The division into even and odd years is supposed to minimize the influence of climate trends on cross-validation results. Switanek et al. (2017, doi:10.5194/hess-21-2649-2017) have demonstrated how such trends can distort cross-validation results if validation and training periods are chosen as suggested by you. Yet I admit that my validation and training datasets are similar in terms of longer-term climate variability. I will add a discussion of this trade-off to the Methods section.

Other minor issues: Page 2, line 16) The author claims to develop a new quantile mapping method that allows for controlled adjustment of biases in all quantiles. As far as I know, most QM bias correction methods do just that. Have I misunderstood something here?

For most variables, the ISIMIP2b bias adjustment method is not a quantile mapping method, see also the description in Section 3.1. To make this clearer, I will write "While structurally different bias adjustment methods (including but not restricted to quantile mapping methods)" instead of "While structurally different bias adjustment methods" in line 16 of page 2.

Page 4 line 26) ". . .tasmax, and tasmin are not adjusted directly. Instead, I adjust biases in pr and prsnratio. . ." The author passes, here and in other parts of the article, from a passive to an active (first person) voice. I suggest he chose one or the other.

I will try to use the passive voice more often in the revised manuscript. I do not see the value in removing all first person voices, though.

Page 7 line 19) x should be $x\_hist^{obs}$, am I right? Also, y needs indexes as well, I think. This should be the case in all the math that follows. I understand that the author is trying to keep the formalism light. I suggest then that the author state that, in what follows, x is actually $x\_hist^{obs}$, etc.

Actually, in the previous line I wrote "let x be a value of $x\_hist^{obs}$", so x is one value whereas $x\_hist^{obs}$ is a time series of many values. This convention is used in the entire Methods sections. To make this clearer I will write "let x be one of the many values of $x\_hist^{obs}$" in the previous line.

Caption figure 4) 3rd Sentence : Panele (c-j) should perhaps be (c-f)?

Right, thank you.

Page 12 line 8/9) Write "instead of conservative" in the place of "and not conservative". Else it is not clear that we are comparing bilinear with conservative methods.

Thank you, will change.

Author's response (black) to comments by Stefan Hagemann (orange)

The author presents a documentation about an improved bias correction technique that separates bias correction and statistical downscaling. The manuscript provides an useful overview on the bias correction method that is supposed to be used in the third phase of the Inter-Sectoral Impact Model Intercomparison Project (ISIMIP). The paper is generally written well, and the describe formalisms are necessary but not too overloaded with equations. I have only a few remarks that should be addressed.

Thank you for this positive general comment.

Section structures should be indicated in the beginning of a section. Currently, in several cases, a description is given where information is missing. The missing information is then provided in the one of the next paragraphs, but in the beginning there was no indication that this is being done. One example for this is sect. 3.2.2., where the detailed description starts on p.11 – line 32 after 5 paragraphs of general description of the downscaling method.

I will add currently missing section structure descriptions to the first paragraph of every section.

Observations used for bias correction are usually available on a fine, regular grid while the coarse GCM grids are usually not regular (e.g. often a Gaussian grid is used). Thus, the usage of the downscaling technique must be clarified for this common case where the fine grid does not fit into the coarse grid. In this respect I was really wondering why it is written on p. 11 – line 7-11 that 'It requires that the coarse grid of the climate simulation data and the fine grid of the climate observation data are compatible ...'

Thank you for this remark. Indeed fine and coarse grid have to be compatible for MBCnSD to be applicable. If they are not, i.e. if there is a fine grid cell that extends across multiple coarse grid cells, then the value of that fine grid cell cannot be determined by independent downscaling of the corresponding coarse grid cell values. So in order to make MBCnSD applicable in such cases as well, the original GCM output first has to be interpolated to a grid that is both compatible with the fine grid and of similar resolution as the the original GCM grid. I will add this clarification to the manuscript.

Even though the manuscript is a documentation on the technical method of the ISIMIP3 bias correction, it should not be concealed that the bias correction cannot compensate for several types of GCM errors, e.g. erroneous shifts of storm tracks in the GCM. In such cases, bias correction may even lead to erroneous climate change signals. With respect to this topic, I suggest pointing towards Maraun, D., T. Shepherd, M. Widmann, G. Zappa, D. Walton, J.M. Gutiérrez, S. Hagemann, I. Richter, P. Soares, A. Hall and L. Mearns (2017). Towards process-informed bias correction of climate change simulations. Nature Clim. Change 7: 764-773, doi:10.1038/nclimate3418.

I did not conceal this caveat but I did and do not think that it needs to be included in this manuscript, given that it is a technical description of one particular method, not a bias adjustment review paper. I am aware of the caveat and I also think that everybody who applies bias adjustment methods should be aware of it but I do not see the need to repeat it in every paper about bias adjustment. This is what I would have to do if I wanted to be consistent and followed your suggestion in this case.

p.3 – line 2: It is written: '... these data are bilinearly interpolated...'

Does 'these data' refers to the original GCM data or to the data interpolated on the 2° grid? I guess you mean the original data as otherwise the bilinear interpolation would not make so much sense as the 0.5° data fit to the 2° data.

Right, this is not clear enough. I will simply remove this sentence because it is only relevant for the LI+BA approach, which is sufficiently elaborated in section 3.3.

p.3 – line 8: As the EWEMBI refers to a published dataset, but not peer-reviewed publication, I suggest including more information about the data, variables (and their respective source) and bias-correction. This may also be done in an appendix.

Thank you. I forgot to mention that all of that information is given in Frieler et al. (2017, http://dx.doi.org/10.5194/gmd-10-4321-2017). I will add the following sentence to the paragraph: " For a description of the EWEMBI dataset including variables covered, data sources used, and bias adjustments applied see Frieler et al. (2017; Sect. 3.1 and Table 1)."

p.5 – Table 2: In table or the associated text, the difference between bounds and threshold should be clearly described. In addition, it should be noted where the different types of trend preservation are prescribed (see also major remark 1).

I agree, thank you. I will add the following sentences to the table caption: "Where a lower (upper) bound is specified, no values less (greater) than this bound will occur in the bias-adjusted data. For every lower (upper) bound, a lower (upper) threshold is defined, which is only slightly greater (less) than the bound. The lower (upper) threshold is used to adjust the relative frequency of values less (greater) than the threshold. […] The different kinds of trend preservation are described in and around equations (1–8)."

p.9 – line 6: Is this the threshold listed in Table 2?

Yes. I will insert "(see Table 2)" after "threshold" to make this clearer.

p.9 – line 7-13: The definitions and differences of a and α as wells as b and β are unclear. Please clarify!

I will replace "Let" in line 6 of page 9 by "For climate variables with a lower bound a and lower threshold α, let", and "bound" in line 10 of page 9 by "bound b and upper threshold β" to clarify this.

p.10 – eq. 10-14: I am not familiar with the logit function, and I assume other readers many not either. Please explain what this function is doing. Or is this just the function you define in eq. 15? This must be indicated when the logit term is used for the first time.

I will append "(see Eq. (15) for the definition of the logit function)" to line 7 of page 10.

p.11 – line 3: It is written: 'Since the resulting data can be considered bias-free ...' The uninformed user may understand this misleading statement wrongly. See also major remark 3.

This statement is misleading, I agree. I will therefore replace "bias-free" by "unbiased in the distribution of daily values per climate variable, grid cell, and calendar month".

p.12 – line 8-10: Sentence "This is done ..." is too long and difficult to read. Please separate into two sentences.

I will chop the sentence in two after "for all k". The then second sentence will be replaced by "Broadcasting with bilinear interpolation is preferred because it results in smoother fields than broadcasting with conservative interpolation, as exemplified in Fig. 6."

p.15 – line 11: The first column is set to W. What about the other columns?

The other columns do not matter as long as O is orthogonal. I will append "(all other columns can be chosen at will)" to the first sentence in this line.

p.17 – line 5 and 8: Replace 'is to represent' by 'represents'.

Will do.

p.19 – Fig. 7 (also Fig. 8, 9, 10): Some of the bluish and greenish colours are difficult to separate. Please improve colour setting.

Will do.

List of changes made to the manuscript (page and line numbers refer to the discussion paper)

- Add outlook to ISIMIP3BASD v2.0 = ISIMIP3BASD v1.0 + multivariate bias adjustment to Conclusions section.
- Add discussion of cross-validation setting to Methods section.
- Write "While structurally different bias adjustment methods (including but not restricted to quantile mapping methods)" instead of "While structurally different bias adjustment methods" in line 16 of page 2.
- Use the passive voice more often.
- Write "Let x be one of the many values of x_hist^obs" instead of "Let x be a value of x_hist^obs" in line 18 of page 7.
- Change "Panels (c-j)" to "Panels (c-f)" in caption of Figure 4.
- Write "instead of conservative" instead of "and not conservative" in line 8/9 of page 12.
- Add currently missing section structure descriptions to the first paragraph of every section.
- Add clarification of how to apply MBCnSD if fine and coarse grids are not compatible.
- Remove sentence "For bias adjustment at 1/2° spatial resolution, these data are bilinearly interpolated to a global 1/2° × 1/2° latitude–longitude grid." from line 1/2 of page 3.
- Add sentence "For a description of the EWEMBI dataset including variables covered, data sources used, and bias adjustments applied see Frieler et al. (2017; Sect. 3.1 and Table 1)." to line 9 of page 3.
- Add "Where a lower (upper) bound is specified, no values less (greater) than this bound will occur in the bias-adjusted data. For every lower (upper) bound, a lower (upper) threshold is defined, which is only slightly greater (less) than the bound. The lower (upper) threshold is used to adjust the relative frequency of values less (greater) than the threshold. […] The different kinds of trend preservation are described in and around equations (1–8)." to the caption of Table 2.
- Insert "(see Table 2)" after "threshold" in line 6 of page 9.
- Replace "Let" in line 6 of page 9 by "For climate variables with a lower bound a and lower threshold α, let", and "bound" in line 10 of page 9 by "bound b and upper threshold β".
- Append "(see Eq. (15) for the definition of the logit function)" to line 7 of page 10.
- Replace "bias-free" by "unbiased in the distribution of daily values per climate variable, grid cell, and calendar month" in line 3 of page 11.
- Chop long sentence in line 8-10 of page 12 in two after "for all k". Replace the then second sentence by "Broadcasting with bilinear interpolation is preferred because it results in smoother fields than broadcasting with conservative interpolation, as exemplified in Fig. 6."
- Append "(all other columns can be chosen at will)" to the first sentence in line 11 of page 15.
- Replace "is to represent" by "represents" in lines 5 and 8 of page 17.
- Use more easily distinguishable colors in figures 7–10.

**Trend-preserving bias adjustment and statistical downscaling with ISIMIP3BASD (v1.0)**

Stefan Lange[1]

[1]Potsdam Institute for Climate Impact Research (PIK), Member of the Leibniz Association, P.O. Box 60 12 03, 14412 Potsdam, Germany

**Correspondence:** Stefan Lange (slange@pik-potsdam.de)

**Abstract.** In this paper I present new methods for bias adjustment and statistical downscaling that are tailored to the requirements of the Inter-Sectoral Impact Model Intercomparison Project (ISIMIP). In comparison to their predecessors, the new methods allow for a more robust bias adjustment of extreme values, preserve trends more accurately across quantiles, and facilitate a clearer separation of bias adjustment and statistical downscaling. The new statistical downscaling method is stochastic and better at adjusting spatial variability than the old interpolation method. Improvements in bias adjustment and trend preservation are demonstrated in a cross-validation framework.

*Copyright statement.* The author agrees to the licence and copyright terms of Copernicus Publications as of 6 June 2017.

[revised manuscript text omitted]
{sim}}_{\text{hist}}$, $x^{\text{sim}}_{\text{fut}}$ to $x^{\text{obs}}_{\text{hist}}$. Let $x^{\text{obs}}_{\text{fut}}$ be the resulting time series of pseudo future observations.

6. (For all variables.) Use parametric quantile mapping to adjust the distribution of values in $x^{\text{sim}}_{\text{fut}}$ to the distribution of values in $x^{\text{obs}}_{\text{fut}}$. For bounded variables, also bias-adjust the frequency of values beyond threshold. Let $y^{\text{sim}}_{\text{fut}}$ be the resulting time series.

7. (For psl, rlds, and tas only.) Add trend subtracted from $x^{\text{sim}}_{\text{fut}}$ in step 3 to $y^{\text{sim}}_{\text{fut}}$.

8. (For rsds only.) Scale values in $y^{\text{sim}}_{\text{fut}}$ back to their actual range.

Steps 1 and 8 are only applied to rsds and reflect that this climate variable has a physical upper bound which varies over the annual cycle. In order to fit this case into the unified framework, which at its core assumes constant bounds and thresholds, rsds values are scaled to the interval $[0, 1]$ in step 1, and back to their actual range in step 8. These scalings are done using annual cycles of upper bounds that are estimated from the rsds values in $x^{\text{obs}}_{\text{hist}}$, $x^{\text{sim}}_{\text{hist}}$, $x^{\text{sim}}_{\text{fut}}$. Following Lange (2018), annual cycles of upper bounds at daily temporal resolution are estimated as running mean values of running maximum values of multi-year daily maximum values. Here, a window length of 31 days is used for the running window calculations. Let $b^{\text{obs}}_{\text{hist}}$, $b^{\text{sim}}_{\text{hist}}$, $b^{\text{sim}}_{\text{fut}}$ be these annual cycles estimated for time series $x^{\text{obs}}_{\text{hist}}$, $x^{\text{sim}}_{\text{hist}}$, $x^{\text{sim}}_{\text{fut}}$, respectively. Let further $x_{ij}$ be the value of one of these time series on day $j$ of year $i$, and let $b_j$ be the upper bound for that day of the year according to the corresponding annual cycle, then $x_{ij} \leq b_j$ holds true for all years $i$ and $j = 1, \ldots, 366$. The scaling in step 1 is done according to $x_{ij} \mapsto x_{ij}/b_j$. The scaling in step 8 requires an annual cycle of upper bounds to the bias-adjusted rsds values. Let $b^{\text{obs}}_{\text{fut}}$ denote this annual cycle. Following Frieler et al. (2017, Eq. (2)), it is estimated according to $b^{\text{obs}}_{\text{fut}} = b^{\text{obs}}_{\text{hist}} b^{\text{sim}}_{\text{fut}} / b^{\text{sim}}_{\text{hist}}$. 
[revised manuscript text omitted]

555